# Incidence of Pertussis in Older Children Underestimated in the Whole-Cell Vaccine Era: A Cross-Sectional Seroprevalence Study

**DOI:** 10.3390/vaccines13020200

**Published:** 2025-02-17

**Authors:** Qian-Qian Du, Qing-Hong Meng, Wei Shi, Kai-Hu Yao

**Affiliations:** 1Key Laboratory of Major Diseases in Children, Ministry of Education, National Clinical Research Center for Respiratory Diseases, National Key Discipline of Pediatrics, Laboratory of Infection and Microbiology, Beijing Pediatric Research Institute, Beijing Children’s Hospital, Capital Medical University, National Center for Children’s Health, Beijing 100045, China; qianqiandu0@126.com (Q.-Q.D.); mengqinghong@aliyun.com (Q.-H.M.); shiwei613115@126.com (W.S.); 2Beijing Children’s Hospital, Capital Medical University, No. 56 Nanlishi Road, Xicheng District, Beijing 100045, China

**Keywords:** pertussis, school-age children, seroepidemiology, immunological response, vaccination, anti-PT IgG

## Abstract

Objectives: China was once a country with a high incidence of pertussis, with reported incidence rates exceeding 100 per 100,000 before the introduction of the pertussis vaccine. After the widespread implementation of the pertussis vaccination program, reported cases of pertussis significantly decreased. This study aimed to investigate the serological prevalence of pertussis among school-age children during the administration of the whole-cell pertussis (wP) vaccine in China. Methods: We selected a representative random sample from different schools, with the inclusion criteria being school-age children without clinical symptoms of pertussis. A total of 368 frozen serum samples were obtained from children aged 6–<18 years at various schools in Guizhou in November 2005 and subsequently analyzed. Results: The positive rate of anti-pertussis toxin (PT) IgG antibodies (>62.5 IU/mL) were 4.9% (16/368) among school-age children. The positive rates of anti-PT IgG antibodies were 3.3%, 3.8%, 4.0%, 3.3%, and 10.8% in children aged 6–<8 y, 8–<10 y, 10–<12 y, 12–<14 y, and 14–<18 y, respectively. The increase in PT-IgG antibody levels among older children was likely due to pertussis infection in these school-age children. The positive rate of anti-PT IgG varied between different schools. The pertussis antibody levels of adolescents aged 14–<18 y were significantly higher than those of school-age children in the younger age group (6–<8 y and 8–<10 y) (*p* = 0.0097 and *p* = 0.0007, respectively). Conclusions: During the era of wP vaccine use, pertussis infections were common among school-age children, particularly in adolescents, with potential unrecognized localized or school-based outbreaks.

## 1. Introduction

Pertussis is a highly contagious respiratory infection caused by the bacterium Bordetella pertussis [1]. Since the implementation of the global Expanded Program on Immunization (EPI) in 1974, there has been a significant reduction in disease incidence [2,3]. From 1 January 2025, the current immunization schedule of administering one dose of the acellular pertussis–diphtheria–tetanus combined vaccine (DTaP) at 3 months, 4 months, 5 months, and 18 months of age and one dose of the diphtheria–tetanus (DT) combined vaccine at 6 years of age will be adjusted to administering one dose of the DTaP vaccine at 2 months, 4 months, 6 months, 18 months, and 6 years of age nationwide [4]. Since 1978, pertussis vaccination has been implemented widely nationwide, achieving and maintaining a high coverage rate of immunization [5]. In 2005, the coverage rate of children receiving at least three doses of the diphtheria–tetanus–pertussis (DTP) vaccine in childcare institutions in Guizhou Province was 90%, and the coverage rate of children receiving at least three doses of the DTP vaccine in primary schools was 88% [6]. In 2007, DTaP was introduced into childhood immunization in China, and it had fully replaced DTwP by 2013 [7]. In 2008, the province of Guizhou began to gradually replace the DTwP vaccine with DTaP for routine immunization of children. The vaccine coverage across China, and particularly in Guizhou, was relatively homogeneous during the study period.

China was once a country with a high incidence of pertussis, with reported incidence rates exceeding 100 per 100,000 before the introduction of the pertussis vaccine. After the widespread implementation of the pertussis vaccination program, reported cases of pertussis significantly decreased, and by the late 1990s, the incidence rate had dropped to less than 1 per 100,000 [8]. From 2006 to 2013, the number of cases nationwide was less than 3000 per year, with fewer than 10 deaths annually [9]. Clinically, patients exhibiting symptoms of pertussis were also rarely seen. Despite efforts to standardize diagnostic tests, there remains some variability in laboratory methods used across different regions, affecting the homogeneity of the surveillance system. Consequently, the reported incidence rate of pertussis in Guizhou has declined significantly, from 58.20/100,000 in 1983 to 2.16/100,000 in 1989, further dropping to 0.269/100,000–1.940/100,000 in 1990–2007. In 2008–2013, the reported incidence rate of pertussis in the province ranged from 0.061/100,000 to 0.221/100,000 [10]. This downward trend has persisted, with the incidence rate remaining below this level since then.

Since the winter of 2023, there has been a marked surge in the number of pertussis cases reported across the country, particularly among preschool- and school-age children [11]. Recently, some community surveillance studies on anti-PT antibodies have indicated that the occurrence of pertussis among older children is notably prevalent and is thousands of times more than the reported figures in China [12]. Is the current outbreak of pertussis among school-age children and adolescents a recently emerged phenomenon or one that has been historically present? A repository of serum samples, collected from school-age and adolescent children and subsequently frozen in 2005, presents a valuable research opportunity for exploring this particular inquiry. To investigate the cause of local outbreaks of nephritis, a total of 368 serum samples were collected from children in eight schools across Rongjiang County and Leishan County in Guizhou, China, in November 2005 [13].

## 2. Materials and Methods

### 2.1. Research Subjects

We selected a representative random sample from different schools across Rongjiang County and Leishan County in Guizhou. None of the serum samples were obtained from children exhibiting symptoms consistent with pertussis. After the initial testing phase, the serum samples were maintained at −80 °C, ensuring they remained in a continuously frozen state until this present test. The amount of each preserved serum exceeded 1 ml in the tube. The patients’ names, ID numbers, residential addresses, and contact information were not collected for this study.

These specimens were obtained from eight schools, comprising 26 specimens from Aixiao Primary School (School A), 39 from Sanjiang Township Primary School (School B), 40 from Balu Primary School (School C), 26 from Guyi Primary School (School D), 84 from Qiaolai Primary School (School E), 43 from Jiahui Primary School (School F), 83 from Kaitun Primary School (School G), and 27 from Mianhuatun Primary School (School H).

### 2.2. Laboratory Testing

The concentration of anti-PT IgG and anti-tetanus toxin (TT) IgG in serum samples was determined using a commercially available ELISA kits (Euroimmun, Lübeck, Germany and Institut Virion/Serion GmbH, Würzburg, Germany, respectively). A level of anti-PT IgG < 5.0 IU/mL was classified as undetectable and defined as seronegative. Levels of anti-PT IgG ≥ 62.5 IU/mL were considered indicative of recent pertussis infection [14]. Levels of anti-TT were categorized as follows: <0.01 IU/mL as seronegative (below basic immunity), 0.01–<0.1 IU/mL as basic immunity, and ≥1 IU/mL as seroprotective (long-term protection) [15].

### 2.3. Statistical Analysis

Data analysis was conducted using GraphPad Prism software (version 8; GraphPad Software, La Jolla, CA, USA) and SPSS (version 26.0). The differences in the proportions or rates between various age groups were compared using the *t*-test. *p* < 0.01 was deemed to indicate statistical significance.

## 3. Results

The positive rate of anti-PT IgG antibodies (>62.5 IU/mL) was 4.9% (16/368). The rates in children aged 6–<8 y, 8–<10 y, 10–<12 y, 12–<14 y, and 14–<18 y were 3.3% (2/61), 3.8% (4/106), 4.0% (3/75), 3.3% (3/89), and 10.8% (4/37), respectively. The pertussis-positive rates in older age groups (14–<18 y) were higher than those in other age groups. The pertussis seroprevalence of each age group was different (Figure 1). It can be seen from the mean value and standard deviation that the pertussis antibodies levels of adolescents aged 14–<18 y were significantly higher than those of school-age children in the younger age groups (6–<8 y and 8–<10 y) (*p* = 0.0097 and *p* = 0.0007, respectively).

The positive rate of anti-PT IgG (>62.5 IU/mL) varied among children attending different schools. Specifically, School D exhibited the highest rate of 19.2%, while School A and School C followed with rates of 11.6% and 10.0%, respectively. On the other hand, the positive rates of pertussis antibodies in the other five schools were significantly lower than that of the total (0.0%~2.5% vs. 4.9%).

The positive rate of anti-TT IgG antibodies gradually decreased with increasing age. The seroprotective rates (>1.0 IU/mL) of anti-TT IgG antibodies were 18.0%, 17.9%, 8.1%, and 5.3% in 6–<8 y, 8–<10 y, 10–<12 y, 12–<14 y, and 14–<18 y, respectively. The protective rate of anti-TT IgG antibodies also varied among different schools. The sequence from highest to lowest, however, deviates notably from the positive rate of anti-PT IgG (Table 1).

## 4. Discussion

It is well known that the duration of immunity in children after vaccination was estimated to wane over 4–12 years [16]. Lambert [17] reported on a 1962 outbreak in Michigan, in which 195 cases of *B. pertussis* infection were identified from 474 household members. Vaccination history was collected, and attack rates were calculated, demonstrating that 95% of the attacks had occurred within 12 years of the last dose of the wP vaccine. By 2005, the standard immunization schedule in China involved administering the final dose of the wP vaccine when children were around 2 years old. This timing was part of a comprehensive vaccination strategy designed to provide robust protection against pertussis during early childhood. Studies have shown that immunity acquired by wP vaccination wanes 6–9 years after the last dose [18,19]. Theoretically, the immunity against pertussis acquired through vaccination should decline with age. However, the proportion of positive results for pertussis antibodies increased with age in this study.

Furthermore, there was an observed clustering of positive cases within schools. This suggested that around 2005, there was a prevalence and even localized outbreaks of pertussis infections among school-age children and adolescents in the local area. Pimentel et al. [20] collected 192 nasopharyngeal swabs for culture and PCR to identify B. pertussis in individuals aged 10 years and over with a cough that had lasted between 14 and 30 days in ten public health outpatient clinics in Brazil. During an interepidemic period, 1 in 20 cases of prolonged cough had pertussis, suggesting that there has indeed been pertussis transmission among adolescents. Before the pandemic, numerous studies involving serological analyses and enhanced clinical surveillance suggested that reported pertussis rates in China likely “substantially underestimated” the true incidence [21].

Importantly, the reported cases proportion of school-age children, adolescents, and adults in China was even lower, at less than 10% of the reported pertussis cases [22]. We compared the results of serological studies with similar age groups in recent years from other regions in China. Liu et al. [23] reported that the positive rate of anti-PT-IgG antibodies was 4.6% (>40.0 IU/mL) in people aged 6–18 y in Youyang Hospital between 2018 and 2019. Studies conducted in Jiangsu reported that the positive rate of anti-PT-IgG antibodies was 7.5% (>40.0 IU/mL) among individuals aged 15–19 y from 2018 to 2021 [24]. The present study in Guizhou estimated a prevalence of pertussis of 4.9% (>62.5 IU/mL) among children aged 6–<18 y and 10.8% (>62.5 IU/mL), and 15.8% (>40.0 IU/mL) among children aged 14–<18 y in 2005. These pieces of evidence collectively suggest that school-age children vaccinated in the wP vaccine era had recurrent B. pertussis infections in China. Because many infections were asymptomatic or resulted in only mild symptoms. They often went unnoticed and undiagnosed. The lack of noticeable symptoms meant that public health authorities and healthcare providers did not recognize the full extent of *B. pertussis* transmission within the schools.

An important limitation of this study is that the samples primarily originate from two relatively remote counties, which limits our ability to generalize the findings to the entire country. However, during that period, the number of reported pertussis cases nationwide was generally low, especially concerning data on preschool- and school-age children and adult patients, which constituted only a small fraction of all reported cases. Otherwise, it does not include a comparative analysis of the effectiveness and outcomes during the era of acellular pertussis vaccines. The study relies on historical data collected during the wP vaccine era, which may not fully represent current conditions. Given this context, it is recommended to enhance monitoring and research efforts focused on pertussis among these age groups to better understand its epidemiological characteristics and develop effective prevention and control strategies.

## 5. Conclusions

During the era of wP vaccine use, pertussis infections were common among school-age children, particularly in adolescents, with potential unrecognized localized or school-based outbreaks. Pertussis has been an ongoing issue in China, particularly among school-age children and older age groups, but it has not received adequate attention.

## Figures and Tables

**Figure 1 vaccines-13-00200-f001:**
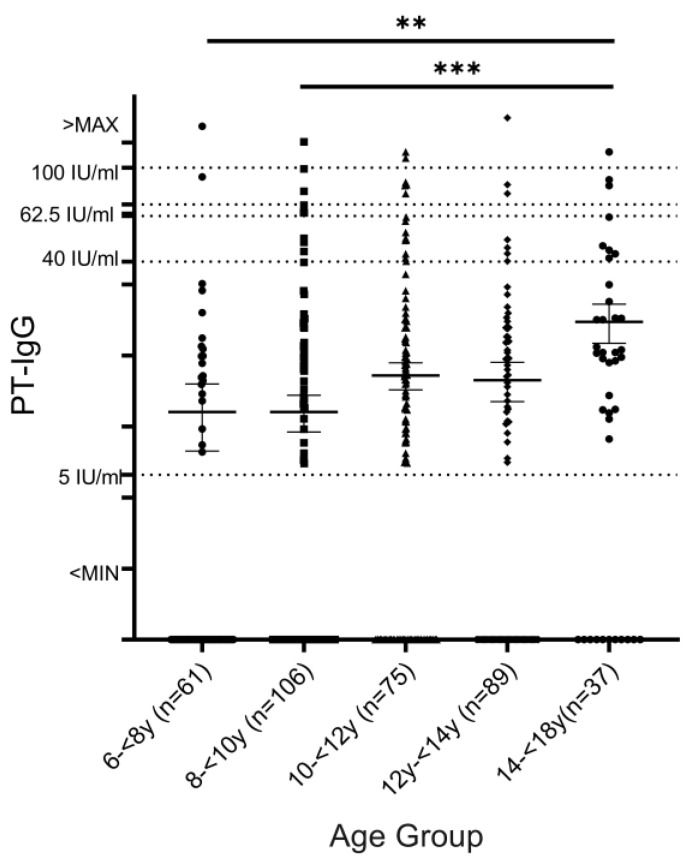
Comparison of anti-PT IgG antibody levels in different age groups in Guizhou, China. Note: *p* < 0.01 was significant difference (**), *p* < 0.001 was extremely significant difference (***).

**Table 1 vaccines-13-00200-t001:** The positivity rate of anti-PT IgG and anti-TT among school-age children in Guizhou, China, 2005.

	No.	Anti-PT IgG			Anti-TT IgG		
		<5 IU/mL	5–62.5 IU/mL	>62.5 IU/mL	<0.1 IU/mL	0.1–1 IU/mL	>1 IU/mL
Total	368	204 (55.4%)	148 (40.2%)	16 (4.9%)	154 (41.8%)	165 (44.8%)	49 (13.4%)
Age							
6–<8 y	61	41 (67.2%)	18 (29.5%)	2 (3.3%)	15 (24.6%)	35 (57.4%)	11 (18.0%)
8–<10 y	106	72 (67.9%)	30 (28.3%)	4 (3.8%)	33 (31.0%)	54 (50.9%)	19 (17.9%)
10–<12 y	75	36 (48.0%)	36 (48.0%)	3 (4.0%)	34 (45.3%)	31 (41.4%)	10 (13.3%)
12–<14 y	89	45 (50.6%)	41 (46.1%)	3 (3.3%)	48 (53.9%)	35 (39.4%)	6 (6.7%)
14–<18 y	37	10 (27.0%)	23 (62.2%)	4 (10.8%)	24 (64.9%)	10 (27.0%)	3 (8.1%)
School							
A	26	16 (61.5%)	7 (26.9%)	3 (11.6%)	8 (30.8%)	16 (61.5%)	2 (7.7%)
B	39	18 (46.2%)	20 (51.3%)	1 (2.5%)	17 (43.6%)	20 (51.28%)	2 (5.1%)
C	40	16 (40.0%)	20 (50.0%)	4 (10.0%)	19 (47.5%)	13 (32.5%)	8 (20.0%)
D	26	4 (15.4%)	17 (65.4%)	5 (19.2%)	10 (38.5%)	9 (34.6%)	7 (26.9%)
E	84	51 (60.7%)	32 (38.1%)	1 (1.2%)	23 (27.3%)	48 (57.1%)	13 (15.6%)
F	43	33 (76.7%)	9 (20.9%)	1 (2.4%)	17 (39.5%)	23 (53.5%)	3 (7.0%)
G	82	43 (52.4%)	38 (46.3%)	1 (1.2%)	47 (57.3%)	24 (29.3%)	11 (12.6%)
H	28	23 (82.1%)	5 (17.9%)	0 (0.0%)	13 (46.4%)	12 (42.9%)	3 (10.7%)

## Data Availability

The datasets used and analyzed during the current study are available from the corresponding author upon reasonable request.

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
