# Peer review of "Incidence of Pertussis in Older Children Underestimated in the Whole-Cell Vaccine Era: A Cross-Sectional Seroprevalence Study"

_vaccines, 2025, doi:10.3390/vaccines13020200_

Round 1
Reviewer 1 Report
Comments and Suggestions for Authors
Pertussis, a vaccine-preventable respiratory disease caused by the Gram-negative bacterium Bordetella pertussis, remains a significant global concern due to its resurgence in many countries. China is no exception to this epidemiological trend. The manuscript presents prevalence data on pertussis among school-age children during the administration of the whole-cell vaccine (wP) in China. A total of 368 frozen serum samples were obtained from children aged 6 to <18 years across various schools in Guizhou in November 2005 and subsequently analyzed. The authors conclude that during the era of whole-cell vaccine use, pertussis infections were common among school-age children. While the manuscript provides data of interest, several aspects require clarification to enable a robust interpretation of the findings and to enhance the scientific rigor of the study. These aspects should be addressed before considering the manuscript for publication.
Specific Comments
- The authors should provide a comprehensive description of pertussis surveillance systems in China, with particular emphasis on Guizhou. Specifically: What laboratory assays were used for pertussis case confirmation? Have these methods evolved over time, and if so, how?Which diagnostic tests were employed in 2005? Clarifying these points is essential for understanding the reliability and comparability of the reported findings.
- While the manuscript mentions vaccine coverage, it is unclear whether the reported coverage refers to the primary 3-dose series or includes booster doses. Additionally: How homogeneous was vaccine coverage across China, and particularly in Guizhou, during the study period? What was the DTP3 coverage in China and Guizhou in 2005?
- Do the authors have access to the vaccination status of the sampled population? Were any of the serum samples obtained from children exhibiting symptoms consistent with pertussis?
- The manuscript should contextualize the findings within the broader epidemiological landscape of pertussis in 2005: What was the overall prevalence of pertussis in China during that year? Were the serum samples collected during an outbreak year? How has the incidence of pertussis evolved over time and across different age groups?
Providing this information would significantly enhance the contextualization of the results.
- To avoid misinterpretations regarding the effectiveness of the whole-cell vaccine, the study should include comparative analyses for the acellular vaccine (aP) era. If such analyses are not feasible, the authors should explicitly state this as a limitation. Otherwise, readers may mistakenly attribute the findings solely to the use of wP.
- Based on the known differences in immunity duration induced by wP and aP vaccines, what would the authors expect to observe with aP use? Addressing this question would add depth to the discussion.
The manuscript should include detailed information on the origin and composition of the pertussis vaccines used during the study period. Specifically: What were the compositions of both the wP and aP vaccines?Who were the manufacturers of these vaccines?
Author Response
Reviewer #1:
Pertussis, a vaccine-preventable respiratory disease caused by the Gram-negative bacterium Bordetella pertussis, remains a significant global concern due to its resurgence in many countries. China is no exception to this epidemiological trend. The manuscript presents prevalence data on pertussis among school-age children during the administration of the whole-cell vaccine (wP) in China. A total of 368 frozen serum samples were obtained from children aged 6 to <18 years across various schools in Guizhou in November 2005 and subsequently analyzed. The authors conclude that during the era of whole-cell vaccine use, pertussis infections were common among school-age children. While the manuscript provides data of interest, several aspects require clarification to enable a robust interpretation of the findings and to enhance the scientific rigor of the study. These aspects should be addressed before considering the manuscript for publication.
Specific Comments:
- The authors should provide a comprehensive description of pertussis surveillance systems in China, with particular emphasis on Guizhou. Specifically: What laboratory assays were used for pertussis case confirmation? Have these methods evolved over time, and if so, how?Which diagnostic tests were employed in 2005? Clarifying these points is essential for understanding the reliability and comparability of the reported findings.
The authors’ answer:
Thank you for your suggestion. Regarding laboratory diagnostic methods for confirming pertussis, the latest Chinese Guidelines for Diagnosis, Treatment, and Prevention of Pertussis (2024 edition) state that a diagnosis can be confirmed if any one of the following pathogen or serological test results is met:
- Isolation of Bordetella pertussis from respiratory specimens;
- Detection of Bordetella pertussis nucleic acid in respiratory specimens;
- A single pertussis toxin (PT) IgG antibody concentration (titer) greater than the recommended threshold for diagnosing acute infection, applicable to children, adolescents, and adults who have been vaccinated with pertussis-containing vaccines more than one year prior;
- A ≥4-fold increase in PT-IgG levels between acute and convalescent sera, suitable for retrospective diagnosis.
These diagnostic methods have been continuously updated over time. In 2005, laboratory serological diagnosis only included the fourth criterion (d). However, in clinical practice, obtaining paired sera during the convalescent phase was difficult, making it challenging to diagnose pertussis promptly. When using serological antibody test results for diagnosis, it is important to consider the course of the disease and the impact of individual vaccination history. This report references the latest recommended thresholds for PT-IgG antibody [Reference 14] and uses WHO-recommended standardized test kits to detect antibodies in serum samples collected from asymptomatic older children and adolescents. The results are reliable and facilitate our ability to retrospectively assess whether pertussis cases were significantly underreported during the whole-cell vaccine era.
- While the manuscript mentions vaccine coverage, it is unclear whether the reported coverage refers to the primary 3-dose series or includes booster doses. Additionally: How homogeneous was vaccine coverage across China, and particularly in Guizhou, during the study period? What was the DTP3 coverage in China and Guizhou in 2005?
The authors’ answer:
Thank you for your suggestion. In the background section, we have further updated the vaccine coverage information in China, as seen in lines 40-45 on pages 1-2. The coverage rates reported in this study include both the primary three doses and full vaccination (≥3 doses). Vaccine coverage was consistent across China, and particularly in Guizhou, during the entire study period. The vaccine coverage across China, and particularly in Guizhou, was relatively homogeneous during the study period. According to the literature:(i). In 2005, the coverage rate of children receiving at least three doses of the diphtheria-tetanus-pertussis (DTP) vaccine in childcare institutions in Guizhou Province was 90%. (ii). In primary schools, the coverage rate for children receiving at least three doses of the DTP vaccine was 88%. In 2005, under the Chinese National Immunization Program (NIP), pertussis vaccination rates were high, with most regions maintaining coverage rates above 90%. This high level of coverage indicates a robust immunization program that aims to protect children against pertussis.
- Do the authors have access to the vaccination status of the sampled population? Were any of the serum samples obtained from children exhibiting symptoms consistent with pertussis?
The authors’ answer:
Thank you for your suggestion. As mentioned in the response to the previous question, we understand that the vaccination rate among school-age children in Guizhou Province in 2005 was approximately 88%. However, we cannot determine the specific vaccination times and doses of the DTP vaccine for the sampled population. We have added a description in the methodology section, as seen in lines 78-80 on page 2. The revised sentence is:
“We selected a representative random sample from different schools across Rongjiang County and Leishan County in Guizhou. There were no serum samples obtained from children exhibiting symptoms consistent with pertussis.”
- The manuscript should contextualize the findings within the broader epidemiological landscape of pertussis in 2005: What was the overall prevalence of pertussis in China during that year? Were the serum samples collected during an outbreak year? How has the incidence of pertussis evolved over time and across different age groups? Providing this information would significantly enhance the contextualization of the results.
The authors’ answer:
Thank you for your suggestion. We have added literature on the epidemiological background of pertussis in China and Guizhou [References 8-10]. Under the surveillance system at that time, it was believed that the overall incidence of pertussis in 2005 did not exceed 1 per 100,000, and no pertussis outbreaks were reported, as seen in lines 50-58 on Page 2. Based on the results of this report, this value appears to have been significantly underestimated. Recently, our research team has published several studies on the evolution of pertussis incidence over time and across different age groups. We have included these references in the discussion.
- To avoid misinterpretations regarding the effectiveness of the whole-cell vaccine, the study should include comparative analyses for the acellular vaccine (aP) era. If such analyses are not feasible, the authors should explicitly state this as a limitation. Otherwise, readers may mistakenly attribute the findings solely to the use of wP.
The authors’ answer:
Thank you for your suggestion. This report does not include a comparative analysis of the acellular vaccine (aP) era. We have added this limitation to the study limitations section, as seen in lines 170-172 on page 5. The revised sentence is:
“Otherwise, it does not include a comparative analysis of the effectiveness and out-comes during the era of acellular pertussis vaccines. The study relies on historical data collected during the whole-cell pertussis vaccine era, which may not fully represent current conditions.”
- Based on the known differences in immunity duration induced by wP and aP vaccines, what would the authors expect to observe with aP use? Addressing this question would add depth to the discussion.
The authors’ answer:
Thank you for your suggestion. We have added literature on the duration of immunity induced by DTwP vaccines [References 17-19] in the discussion section. The primary purpose of including this information is to explain that the elevated pertussis serum IgG levels reported in this study are due to recent acute infections. Additionally, higher serum antibody levels were observed in older age groups, which can be attributed to the gradual waning of vaccine-induced immunity with increasing age. Our research team has published several studies on the surge in pertussis cases in China, many of which were observed during the use of acellular pertussis (aP) vaccines [References11, 22]. However, since this report does not include a comparative analysis of the acellular vaccine era, we have not expanded the discussion on this aspect.
- The manuscript should include detailed information on the origin and composition of the pertussis vaccines used during the study period. Specifically: What were the compositions of both the wP and aP vaccines?Who were the manufacturers of these vaccines?
The authors’ answer:
Thank you for your suggestion. Whole-cell pertussis (wP) vaccines are formulated from whole-cell pertussis bacteria and contain various components, including effective antigens as well as harmful substances like lipopolysaccharides that can cause adverse reactions. Post-vaccination side effects with wP vaccines are relatively frequent and can be more severe. In China, three types of vaccines containing pertussis components are currently available:
(i.) Adsorbed acellular DTP (diphtheria-tetanus-pertussis) combined vaccine;
(ii.) Adsorbed acellular DTP-inactivated polio and Haemophilus influenzae type b (Hib) conjugate combined vaccine (pentavalent vaccine);
(iii.) Acellular DTP-Hib combined vaccine (quadrivalent vaccine).
The vaccine included in China's National Immunization Program (NIP) is the adsorbed acellular DTP vaccine. This vaccine is made from acellular pertussis vaccine antigen, diphtheria toxoid antigen, and tetanus toxoid antigen, with added aluminum hydroxide adjuvant. Manufacturers include the Wuhan Institute of Biological Products, Lanzhou Institute of Biological Products, and others.
This particularly brief report is a sort of "least publishable unit paper" but the message is of interest, albeit not being really original.
Reviewer 2 Report
Comments and Suggestions for Authors
This particularly brief report is a sort of "least publishable unit paper" but the message is of interest, albeit not being really original.
Major issue:
The authors need to include references and discussion of previous reports that showed that B. pertussis was circulating in the wP vaccine ara, such as:
Grimprel E, Bégué P, Anjak I, Njamkepo E, François P, Guiso N. Long-term
human serum antibody responses after immunization with whole-cell pertussis
vaccine in France. Clin Diagn Lab Immunol. 1996 Jan;3(1):93-7. doi:
10.1128/cdli.3.1.93-97.1996. PMID: 8770511
MInor issues:
The reviewer disagrees with the statement on l. 141-143 There is no doubt that schooled children were repeatedly infected with B. pertussis in the whole cell pertussis vaccine era, as also documented int he paper by Grimprel et al. cited above. Those infections were largely asymptomatic, however, which is the reason why attention was not paid to them and whooping cough was not diagnosed in those iB. pertussis-infected children. The situation has dramatically changed within 7 - 10 years form introduction of the acellular pertussis vaccines, where bona fide whooping outbreaks started to occur and mostly teenager and adolescent patients started to seek medical help, and thus were diagnosed with long-lasting characteristic paroxysmal whooping cough symptoms in Australia (2009-2011), California (2010 and 2014), EU (2012). Most recently, unprecedentedly massive whooping cough outbreaks swept through the EU, as in 2023 in Denmark, Serbia or Croatia and in 2024 in Czechia, Netherlands, Austria, France and UK. In most EU countries using the aP vaccine, these epidemics reaches a clinically diagnosed whooping cough incidence of over 1,000/100,000 in the highly aP vaccinated and boosted cohorts of 10 - 19 years old ... This is unprecedented.
The authors thus need to look into those data and revise the statement on lines 141-143, as the B. pertussis transmission in the wP vaccine era was more-or-less asymptomatic, which is why it escaped attention...
Author Response
Reviewer #2:
Comment 1:Major issue:
The authors need to include references and discussion of previous reports that showed that B. pertussis was circulating in the wP vaccine ara, such as:
Grimprel E, Bégué P, Anjak I, Njamkepo E, François P, Guiso N. Long-term human serum antibody responses after immunization with whole-cell pertussis vaccine in France. Clin Diagn Lab Immunol. 1996 Jan;3(1):93-7. doi:10.1128/cdli.3.1.93-97.1996. PMID: 8770511
The authors’ answer:
Thank you for your suggestion, we have included the new reference 18 with Grimprel, et al, 1996, Clin Diagn Lab Immunol. This part has been added to the ‘Discussion’ according to your suggestion (lines 132-140 on page 5).
The added section:
“…Lambert [17] reported on a 1962 outbreak in Michigan, in which 195 cases of B. pertus-sis infection were identified from 474 household members. Vaccination history was collected, and attack rates were calculated, demonstrating that 95% of the attacks had occurred within 12 years since the last dose of wP vaccine. By 2005, the standard immunization schedule in China involved administering the final dose of the whole-cell pertussis vaccine when children were around 2 years old. This timing was part of a comprehensive vaccination strategy designed to provide robust protection against pertussis during early childhood. Studies have shown that immunity acquired by wP vaccination wanes 6–9 years after the last dose [18,19].”
Comment 2:Minor issues:
The reviewer disagrees with the statement on l. 141-143 There is no doubt that schooled children were repeatedly infected with B. pertussis in the whole cell pertussis vaccine era, as also documented in the paper by Grimprel et al. cited above. Those infections were largely asymptomatic, however, which is the reason why attention was not paid to them and whooping cough was not diagnosed in those iB. pertussis-infected children. The situation has dramatically changed within 7 - 10 years form introduction of the acellular pertussis vaccines, where bona fide whooping outbreaks started to occur and mostly teenager and adolescent patients started to seek medical help, and thus were diagnosed with long-lasting characteristic paroxysmal whooping cough symptoms in Australia (2009-2011), California (2010 and 2014), EU (2012). Most recently, unprecedentedly massive whooping cough outbreaks swept through the EU, as in 2023 in Denmark, Serbia or Croatia and in 2024 in Czechia, Netherlands, Austria, France and UK. In most EU countries using the aP vaccine, these epidemics reaches a clinically diagnosed whooping cough incidence of over 1,000/100,000 in the highly aP vaccinated and boosted cohorts of 10 - 19 years old ... This is unprecedented.
The authors thus need to look into those data and revise the statement on lines 141-143, as the B. pertussis transmission in the wP vaccine era was more-or-less asymptomatic, which is why it escaped attention...
The authors’ answer:
Thank you very much for your detailed suggestions. The data you provided are important, and we have revised this section as requested to avoid any ambiguity. Your feedback has helped us improve the clarity and precision of our report, ensuring that the information is presented most accurately and understandably as possible. It has been modified according to your suggestion (lines 161-167 on page 5). The revised line is:
“These pieces of evidence collectively suggest that during the whole-cell pertussis vaccine era, school-age children experienced repeated infections with B. pertussis in China. Because many infections were asymptomatic or resulted in only mild symptoms. They often went unnoticed and undiagnosed. The lack of noticeable symptoms meant that public health authorities and healthcare providers did not recognize the full extent of B. pertussis transmission within the schools.”
Reviewer 3 Report
Comments and Suggestions for Authors
This brief report describes the seroprevalence of specific antibodies to pertussis and tetanus toxins in historical samples retrieved from the pediatric population and is relevant for evaluation of vaccination program's efficacy and understanding of epidemiological situation. Below are a few remarks for the authors:
The authors mention a large underestimation of pertussis infections but do not provide any numbers. It would be interesting to know the reported occurence of pertussis for the sampling period to understand what was the registered cases rate.
It would be also good to be more precise in describing the population of this study, the outbreak of nephritis is mentioned however it is unclear whether the children were selected according to the presence of symptoms or do they represent a random population sample, what were the enrollment conditions and what is known about clinical characterization and, if available, the vaccination status of the study participants?
The information of the local ethical assessment of the human sera analysis is missing, please kindly provide it.
The figure seems to be of rather low resolution, please make sure the high-quality graph is available.
Author Response
Reviewer #3:
Comment 1: This brief report describes the seroprevalence of specific antibodies to pertussis and tetanus toxins in historical samples retrieved from the pediatric population and is relevant for evaluation of vaccination program's efficacy and understanding of epidemiological situation. Below are a few remarks for the authors:
The authors’ answer 1 :
Thank you for your positive feedback.
Comment 2: The authors mention a large underestimation of pertussis infections but do not provide any numbers. It would be interesting to know the reported occurence of pertussis for the sampling period to understand what was the registered cases rate.
The authors’ answer 2 :
In response to your feedback, we have expanded the background section to include comprehensive information on how pertussis incidence has changed over time in both China and Guizhou. This additional context provides a clearer picture of the epidemiological trends and patterns in these regions (lines 36-40 on page 1, lines 50-58 on page 2).
Comment 3: It would be also good to be more precise in describing the population of this study, the outbreak of nephritis is mentioned however it is unclear whether the children were selected according to the presence of symptoms or do they represent a random population sample, what were the enrollment conditions and what is known about clinical characterization and, if available, the vaccination status of the study participants?
The authors’ answer 3 :
Thank you for your suggestion. We selected a representative random sample from different schools, with the inclusion criteria being school-age children without clinical symptoms of pertussis. We have provided a more detailed description of the study population in the methods section, as seen in lines 78-80 on page 2. However, we cannot determine the specific vaccination times and doses of the DTP vaccine for the sampled population.
Comment 4: The information of the local ethical assessment of the human sera analysis is missing, please kindly provide it.
The authors’ answer 4 :
Thank you for your suggestion. The patients’ name, ID number, residential address and contact information were not collected for this study. The source of the serum used in this study can be found in reference 13. Ethical issues are not a concern in this case, as there were no ethical requirements for research at that time, so there is no need to specify them particularly.
Comment 5: The figure seems to be of rather low resolution, please make sure the high-quality graph is available.
The authors’ answer 5 :
Thank you for your suggestion. To ensure better clarity and detail, we have replaced the previous images with higher-resolution versions (1200 dpi).
Round 2
Reviewer 1 Report
Comments and Suggestions for Authors
The authors have adequately addressed all comments